# Effect of Mold Width on the Flow Field in a Slab Continuous-Casting Mold with High-Temperature Velocity Measurement and Numerical Simulation

**Jian-Qiu Liu [1], Jian Yang [1,*], Chao Ma [1], Yi Guo [1], Wen-Yuan He [2], Chang-Liang Zhao [2], Ren-Bo Jiang [3] and Yin-Tao Guo [3]**

[1]  State Key Laboratory of Advanced Special Steel, School of Material Science and Engineering, Shanghai University, Shanghai 200444, China; liujianqiu@shu.edu.cn (J.-Q.L.); mc1457769184@shu.edu.cn (C.M.); guoyi@shu.edu.cn (Y.G.)

[2]  Steelmaking Department, Shougang Jingtang United Iron and Steel Co., Ltd., Tangshan 063200, China; hewy2012@sgjtsteel.com (W.-Y.H.); zhaocl5270@sgjtsteel.com (C.-L.Z.)

[3]  Steelmaking Department, Tangshan Stainless Steel Co., Ltd., Tangshan 063000, China; jiangrenbo@hbisco.com (R.-B.J.); gyt3133@163.com (Y.-T.G.)

*  Correspondence: yang_jian@t.shu.edu.cn; Tel.: +86-21-6613-6580

**Abstract:** In this paper, the effects of the width of the mold on the surface velocity, flow field pattern, turbulent kinetic energy distribution, and surface-level fluctuation in the mold were studied with measurement of the flow velocity near the surface of the mold at high temperature with the rod deflection method and numerical calculation with the standard $k$-$\varepsilon$ model coupled with the discrete-phase model (DPM) model for automobile exposed panel production. Under the conditions of low fixed steel throughput of 2.2 ton/min, a nozzle immersion depth of 140 mm, and an argon gas flow rate of 4 L/min, as the width of the mold increases from 880 mm to 1050 mm and 1300 mm, the flow velocity near the surface of the mold decreases. The flow direction changes from the positive velocity with the mold widths of 880 mm and 1050 mm to the unstable velocity with the mold width of 1300 mm. The calculated results are in good agreement with the measured results. The turbulent kinetic energy near the submerged entry nozzle (SEN) gradually increases, and the risk of slag entrainment increases. Under the conditions of high fixed steel throughput of 3.5 ton/min, the SEN immersion depth of 160 mm, and the argon gas flow rate of 10 L/min, as the width of the mold increases from 1600 mm to 1800 mm and 2000 mm, the velocity near the mold surface decreases. The flow velocity at 1/4 of the surface of the mold is positive with the mold width of 1600 mm, while the velocities are negative with the widths of 1800 mm and 2000 mm. The calculated results are basically consistent with the measured results. The high turbulent kinetic energy area near the nozzle expands to a narrow wall, and the risk of slag entrainment is significantly increased. In both cases of low and high fixed steel throughput, the change rules of the flow field in the mold with the width are basically the same. The argon gas flow rate and the immersion depth of SEN should be adjusted reasonably to optimize the flow field in the mold with different widths under the same fixed steel throughput in the practical production.

**Keywords:** mold width; flow field in mold; high-temperature measurement; numerical simulation; surface velocity

## 1. Introduction

In the production process of automobile exposed panels, improper control of the flow field in the slab continuous-casting mold tends to bring about mold flux entrainment, large inclusions, and bubble-type inclusions in the continuous-casting slab, which may result in linear defects on the hot-dip-galvanized automobile exposed panel, giving rise to serious surface quality problems [1]. The continuous-casting process parameters, such as casting speed, argon gas flow rate, immersion depth, and nozzle structure of the submerged

entry nozzle (SEN), can be usually adjusted with the results of mathematical simulation and water modeling to optimize the mold flow field for improving the surface quality of automobile exposed panels.

In recent years, the size and distribution of bubbles in liquid steel and the influence of bubble size on gas–liquid two-phase flow has been widely studied. Li [2] used the 1:4 water model to investigate the two-phase flow characteristics and the bubble size distribution. A mathematical model based on the Euler–Euler approach was developed to analyze the bubble aggregation and breakage in the bubbly flow. Liu [3] used the population balance equations combined with the Eulerian–Eulerian two-phase model to predict the polydispersed bubbly flow inside the slab continuous-casting mold, and the effects of the water flow rate and the gas flow rate on the average bubble size were studied. Cho [4] used the 1:3 water model with the aid of a high-speed video camera and analytical models to investigate the gas pressure, initial bubble size, bubble descending velocity, bubble residence time, and bubble size distribution. Meanwhile, mathematical modeling has been developed to accurately simulate the mechanism of various defects, such as inclusions and the formation of oscillation marks [5–10]. Wu [7,8] considered the evolution of the solid shell, including a fully solidified strand and a partially solidified dendritic mushy zone, which strongly interacts with the turbulent flow and in the meantime is subject to continuous deformation due to the funnel-type mold. Liu [9] developed a Euler–Euler model to predict the argon-steel-slag three-phase flow and the formation of the exposed slag eye in a slab continuous-casting mold.

With regard to the influence of the casting speed, the argon gas flow rate and the immersion depth of the SEN on the mold flow field, Zhang [11] used numerical simulation to study the fluctuation of molten steel on the mold surface at different casting speeds. When the casting speed is low, the change in casting speed has little effect on the fluctuation of the steel-slag liquid level. However, when the casting speed is high, a small change in casting speed will aggravate the liquid-level fluctuation. In addition, when the casting speed is changed uniformly, the surface fluctuation of the liquid level is small. Salas [12] and Zhang [13] researched the effect of SEN immersion depth on the flow field pattern, free-surface fluctuation, and slag layer exposure in the mold by combining water modeling and numerical simulation. With increasing immersion depth, the surface velocity of molten steel, the fluctuation range of the free surface, and the exposed area of the slag layer decrease, but the number of bubbles and the fluctuation of liquid surface near the SEN increase.

In view of the influence of the SEN structure on the flow field of the mold, Cho [14] found that when the nozzle inclination angle is 15° downward, the flow field in the mold is a typical single-roll flow (SRF), the liquid level fluctuates sharply, and the risk of mold flux entrapment increases. When the nozzle inclination angle is 30° downward, the flow field in the mold is a double-roll flow (DRF). Lee [15] reported that for the nozzle downward inclination in the range of 0°–15°, the flow velocity in the upper circulation area is unstable and the liquid level of the mold fluctuates sharply. When the downward inclination angle is greater than 20°, the flow velocity on the surface of the mold decreases and the fluctuation range of the mold liquid level decreases. When argon gas is injected, owing to the buoyancy of the gas bubbles, the unstable range of the nozzle inclination is extended to 20°. Ismael [16,17] indicated that under the same conditions, compared with a square nozzle, the turbulent jet formed by a round nozzle increases, and the risk of mold flux entrainment increases from the results of the large eddy simulation (LES) model and water modeling.

Generally, since the width of the slab mold used for different widths of automobile exposed panels is also different, it is necessary to optimize the mold flow fields of different widths to improve the surface quality of automobile exposed panels with various widths. In view of the characteristics of the flow fields in the mold with different widths, in our recent works [18–20], the flow fields of slab molds with narrow, medium, and wide widths have been studied through rod deflection high-temperature measurement combined with

numerical simulation. For the narrow width [18], increasing casting speed increases the subsurface velocity of molten steel and shifts down the position of the vortex core in the downward circulation zone. For the medium width [19], when the casting speed is increased from 1.0 to 1.3, 1.5, and 2.0 m/min, the flow pattern in the mold changes from SRF to unstable flow (UF) and then to DRF. In the case of wide width [20], when the argon gas flow rate is increased from 6 L/min to 10 L/min and 14 L/min, the flow pattern changes from DRF to SRF, and the risk of slag entrainment increases. With the casting speed increased from 1.0 m/min to 1.1 m/min, the flow field pattern changes from SRF to strong DRF.

With regard to the influence of the width on the flow field of the mold, the research methods can be divided into two kinds. One is fixing the casting speed. Ma [21] found that the larger is the mold width, the greater is the impact pressure of the molten steel stream on the narrow wall, and the greater is the turbulent kinetic energy on the mold liquid surface. Zhang [22] concluded that under the conditions of the same casting speed and other parameters, the liquid-level fluctuation in the narrow-width mold is severer than that of the wide-width mold and the impact pressure of the liquid steel stream on the narrow wall is also larger. Tang [23] reported that with increasing mold width from 1400 mm to 1600 mm and 2150 mm under the same casting speed, the surface velocity gradually decreases and the point where the molten steel stream hits the narrow wall gradually moves downward. The second research method is fixing the steel throughput. Deng [24] found that when the steel throughput is 3.6 ton/min, increasing the mold width can reduce the average wave height of the liquid level and decrease the surface flow velocity to reduce the incidence of mold flux shearing entrainment.

The flow velocity of molten steel on the surface of the mold has an important influence on the surface quality of the automobile exposed panel. When the surface velocity of the steel–slag interface is greater than the critical velocity of slag entrainment, shearing instability (Kelvin–Helmholtz instability [25]) will occur on the steel–slag interface, resulting in shearing entrainment. At the same time, too low surface flow velocity will cause uneven slag consumption due to excessively low molten steel surface temperature and poor slagging. Therefore, it is necessary to know the flow velocity on the surface of the mold. However, it is difficult to measure the surface velocity of molten steel at high temperature. In early research works, the methods for measuring the velocity of molten steel on the surface of the mold [26–28] included flow control sensors, Karman eddy current probes, and sub-meniscus velocity control sensors. Subsequently, the nail-dipping method [25,29] was used to measure the surface velocity of molten steel. Szekely [30] developed a mechanical force reaction probe method to measure the velocity in a turbulent, electromagnetically driven recirculating low-melting-alloy system. A 19-mm-diameter non-ferromagnetic stainless steel disc was fastened to the end of a spring-loaded rod. The end of the rod was connected to a Hewlett-Packard model 7DC-3000 Linear Voltage Differential Transformer (Hewlett-Packard Company, Palo Alto, CA, USA). The pressure on the disc was converted to the (stagnation) velocity. To measure the near-surface velocity of the liquid metal, a drag-form strain gauge system [31] was also developed at MEFOS (Swedish National Metallurgical Research Institute).

In recent years, we and others [32] developed the rod deflection method. When the velocity-measuring rod is inserted into molten steel in the mold to reach a stable deflection under gravity, buoyancy, and the impact force of molten steel, the deflection angle is read and the surface velocity of molten steel can be calculated. The method can accurately and conveniently measure the flow velocity near the surface of molten steel at high temperature.

The flow in a continuous-casting mold is highly turbulent and extremely complex. The geometry and a number of process parameters exert a considerable influence on the flow behavior. Since the surface quality of the automobile exposed panel is highly dependent on the flow field in the mold, an in-depth understanding of these relationships is important for optimization and effective control of the process. For this reason, a large number of studies have already been carried out on this subject in the past. However, under fixed steel throughput, no papers reported the effect of mold width on the mold flow field assisted with the quantitative measurement of mold surface velocity at high temperature, which is beneficial not only for improving the industrial operation of continuous casting but also obtaining knowledge for the scientific community.

In this paper, the effect of mold width on the flow field in the mold is studied at fixed steel throughputs of 2.2 ton/min and 3.5 ton/min. The rod deflection method is used to measure velocity near the mold surface for mold widths of 880 mm, 1050 mm, 1300 mm, 1600 mm, 1800 mm, and 2000 mm. The numerical calculation is conducted by use of the standard $k$-$\varepsilon$ model coupled with the discrete-phase model (dpm) model, and the calculation and measurement results are compared and analyzed. The influence of mold width on the surface velocity, flow field pattern, turbulent kinetic energy, and surface fluctuation of molten steel in the mold is clarified, which will provide technical guidance for improving the surface quality of automobile exposed panels with different widths.

## 2. High-Temperature Velocity Measurement Method

The high-temperature velocity measurement results were obtained by the rod deflection method in the steel plant of Shougang Jingtang Iron & Steel Co., Ltd. (Tangshan, China) and Tangshan Stainless Steel Co., Ltd. in China. As shown in Figure 1, the speed-measuring device of the rod deflection method is composed of a balance block, a deflection bearing, a deflection angle indicator, and a stainless steel detecting rod. The function of the balance block is to make the center of gravity of the entire speed-measuring device fall near the center of the deflection axis, which enables the subtle changes in the velocity of the molten steel to be sensitively detected by the change in the deflection angle of the detecting rod. When measuring the flow velocity near the surface of the mold, the stainless steel detecting rod is inserted into the place near 1/4 of the wide surface of the mold, and the flowing molten steel deflects the detecting rod at a certain angle. By use of the measured deflection angle combined with Equation (1), the surface velocity of the mold can be calculated [32].

$$U_0 = \sqrt{\frac{2(GL_1 \tan\theta - FL_2 \tan\theta)}{L_2 C_D \rho A}} \tag{1}$$

where $U_0$ is the velocity near the mold surface (m/s). $G$ is the gravity of the detecting rod (N). $L_1$ is the gravity arm (m). $F$ is the buoyancy (N). $L_2$ is the buoyancy arm (m). $C_D$ is the resistance coefficient of the flow. $\rho$ is the density of molten steel (kg/m$^3$). $A$ is the projected area of the immersion part of the detecting rod in the direction vertical to the flow of molten steel (m$^2$). The detailed description of the rod deflection method and the error discussion are available in our previous paper [32].

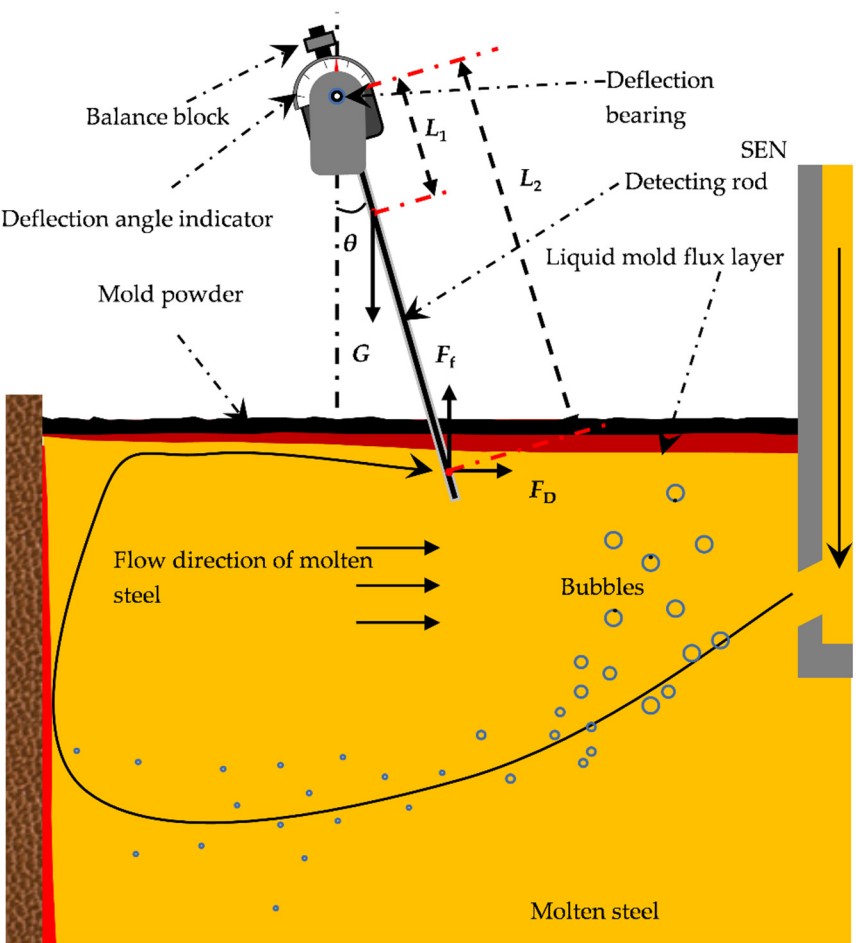

**Figure 1.** Schematic diagram of measuring velocity near the mold surface by the rod deflection method.

### 3. Mathematical Model

*3.1. Numerical Simulation*

In this paper, the standard *k-ε* model and the DPM model were used to simulate the turbulent flow in the submerged entry nozzle and continuous-casting mold. To simplify the complex solution process, the following assumptions were made. Molten steel is an incompressible fluid. Argon gas bubbles are all rigid spheres with uniform diameter and are not affected by pressure and temperature, ignoring the merging and rupture of bubbles. The discrete phase did not occupy the volume in the calculation domain and was regarded as a mass point. The influence of the solidified shell on the flow field was ignored. The influence of mold flux on liquid-level fluctuation and flow field was ignored.

*3.2. Governing Equation*

In the calculation of the two-phase flow of molten steel and argon gas bubbles in the mold, the continuous phase of molten steel was calculated using the Euler method, the discrete phase of argon gas bubbles was calculated using the Lagrangian method, and the discrete and continuous phases were coupled through the source term in the governing equation. There was bidirectional coupling between the continuous phase and the discrete phase. Namely, there was an interaction between the bubbles and the continuous phase, and the movement of the bubbles was affected by the continuous phase. Conversely, the bubbles also had an impact on the movement of the continuous phase.

The movement of the continuous phase is described by Equation (2) of the conservation of mass and Equation (3) of the conservation of momentum.

$$\frac{\partial}{\partial t}(\alpha_1 \rho_1) + \nabla \cdot (\alpha_1 \rho_1 \vec{v}_1) = 0 \tag{2}$$

$$\frac{\partial}{\partial t}\left(\alpha_1\rho_1\vec{v}_1\right) + \vec{v}_1\cdot\nabla(\alpha_1\rho_1\vec{v}_1) = -\nabla p + \nabla\cdot[\alpha_1(\mu_1+\mu_t)]\nabla\vec{v}_1 + \vec{F}_k \tag{3}$$

where $\vec{v}_1$ is the flow velocity (m/s). $\rho_1$ is the density of molten steel (kg/m$^3$). $p$ is the pressure (N/m$^2$). $\mu_1$ is the liquid viscosity (Pa·s). $\mu_t$ is the turbulent viscosity (Pa·s). $\mu_1+\mu_t$ is the effective viscosity (Pa·s). $g$ is the gravitational acceleration (m/s$^2$). $F_k$ is the force of argon gas bubbles acting on molten steel (N/m$^3$). $\alpha_1$ is the liquid-phase volume fraction, which is expressed as

$$\alpha_1 = 1 - \frac{\sum_i V_{d,i}}{V_{cell}} \tag{4}$$

where $V_{d,i}$ is the discrete-phase volume (m$^3$). $V_{cell}$ is the volume of the grid cell (m$^3$).

The standard $k$-$\varepsilon$ model was used to simulate turbulent flow, where the turbulent viscosity $\mu_t$ of the liquid could be expressed as a function of turbulent kinetic energy of $k$ and turbulent energy dissipation rate of $\varepsilon$.

$$\mu_t = C_\mu\rho_1\frac{k^2}{\varepsilon} \tag{5}$$

The influence of molecular viscosity was ignored in the standard $k$-$\varepsilon$ model, and at the same time, a model of transmission equation for $k$ and $\varepsilon$ was established as follows:

$$\alpha_1\rho_1(\frac{\partial k}{\partial t} + \vec{u}_1\cdot\nabla k) = -\nabla(\alpha_1\frac{\mu_t}{\sigma_k}\nabla k) + \alpha_1 G_k - \alpha_1\rho_1\varepsilon \tag{6}$$

$$\alpha_1\rho_1(\frac{\partial\varepsilon}{\partial t} + \vec{u}_1\cdot\nabla\varepsilon) = -\nabla(\alpha_1\frac{\mu_t}{\sigma_\varepsilon}\nabla\varepsilon) + \alpha_1 C_1\frac{\varepsilon}{k}G_k - \alpha_1 C_2\rho_1\frac{\varepsilon^2}{k} \tag{7}$$

where the constants in the model are $C_\mu = 0.09$, $\sigma_k = 1.00$, $\sigma_\varepsilon = 1.30$, $C_1 = 1.44$, and $C_2 = 1.92$ [13].

$G_k$ is the amount of turbulent kinetic energy generated, which is expressed as

$$G_k = \mu_t\left(\frac{\partial u_{i,j}}{\partial x_j} + \frac{\partial u_{i,j}}{\partial x_i}\right)\frac{\partial u_{i,j}}{\partial x_j} \tag{8}$$

In this paper, the discrete-phase model was used to calculate the influence of argon bubbles on the flow field in the mold. The discrete-phase model is a multiphase flow model that tracks the dispersed phase in the Lagrangian coordinate system. Newton's second law is used to calculate the movement of bubbles in the mold [33].

$$m_d\frac{d\vec{v}_d}{dt} = \vec{F}_d + \vec{F}_p + \vec{F}_b + \vec{F}_{vm} + \vec{F}_g + \vec{F}_L \tag{9}$$

where $m_d$ is the bubble mass (kg). $\vec{v}_d$ is the bubble velocity (m/s). $\vec{F}_d$ is the drag force (N). $\vec{F}_p$ is the pressure (N). $\vec{F}_b$ is the buoyancy force (N). $\vec{F}_{vm}$ is the virtual mass force (N). $\vec{F}_g$ is gravity (N). $\vec{F}_L$ is the lift force (N).

$\vec{F}_d$ represents drag force and is defined as

$$\vec{F}_d = C_d\frac{\rho_1\left|\vec{v}-\vec{v}_d\right|\left(\vec{v}-\vec{v}_d\right)}{2}\frac{\pi d_d^2}{4} \tag{10}$$

where the drag coefficient $C_d$ is defined as $C_d = \frac{24}{Re_d}$ when $Re_d < 2$, $C_d = \frac{24}{Re_d}(1 + 0.15Re_d^{0.687})$ when $2 < Re_d < 500$, and $C_d \approx 0.44$ when $Re > 500$, where $Re_d$ is defined as $Re_d = \frac{\rho_f d_d\left|\vec{v}-\vec{v}_d\right|}{\mu_1}$.

$\vec{F}_p$ is the pressure gradient force and defined as

$$\vec{F}_p = \frac{1}{6}\pi d_d^3\rho_1\frac{d\vec{v}}{dt} \tag{11}$$

$\vec{F}_b$ is buoyancy, which can be defined as

$$\vec{F}_{\mathrm{b}} = -\frac{1}{6}\pi d_{\mathrm{d}}^3 \rho_{\mathrm{l}} \cdot \vec{g} \qquad (12)$$

The virtual mass force of $\vec{F}_{\mathrm{vm}}$ is related to the acceleration of the liquid around the bubble, which can be defined as

$$\vec{F}_{\mathrm{vm}} = \frac{1}{6}\pi d_{\mathrm{d}}^3 C_{\mathrm{vm}} \rho_{\mathrm{f}} \frac{d}{dt}\left(\vec{v} - \vec{v}_{\mathrm{d}}\right) \qquad (13)$$

Among them, for spherical objects moving in the liquid phase, $C_{\mathrm{vm}} = 0.5$.

$\vec{F}_{\mathrm{g}}$ is the gravity force (N) and is defined as

$$\vec{F}_{\mathrm{g}} = \frac{1}{6}\pi d_{\mathrm{d}}^3 \rho_{\mathrm{d}} \cdot \vec{g} \qquad (14)$$

Owing to the horizontal velocity gradient, when a bubble rises in the liquid, a lateral lift force of $\vec{F}_{\mathrm{L}}$ (N) acts on it, which is defined as

$$\vec{F}_{\mathrm{L}} = -\frac{1}{6}\pi d_{\mathrm{d}}^3 C_{\mathrm{L}} \rho_{\mathrm{l}} \left(\vec{v}_{\mathrm{d}} - \vec{v}\right)\left(\nabla \cdot \vec{v}\right) \qquad (15)$$

$C_{\mathrm{L}}$ is the lift coefficient, $\lim\limits_{x \to \infty} C_{L} = 0.5$.

The random walk model was used to consider the influence of turbulent velocity fluctuation on bubble motion. The trajectory of the bubble is determined as

$$\vec{x}_{d,i} = \int \vec{v}_{d,i} dt \qquad (16)$$

### 3.3. Calculation Area and Boundary Conditions

Figure 2a is a schematic diagram of the calculation domain and grid division, including the velocity inlet, the pressure outlet, the immersion nozzle, and the entire calculation domain in the mold. The calculation domain was divided into 540,000 cells, and the mesh sizes at the immersion nozzle were refined in particular. The core part of the fluid flow was calculated more accurately to improve the reliability of the calculation results, as shown in Figure 2b. The maximum grid size of the immersion nozzle was 16 mm, and the maximum grid size of the remaining parts was 32 mm. The specific experiment and simulation parameters of each part are listed in Table 1. The chemical compositions of the steel grade involved in the numerical simulation are shown in Table 2.

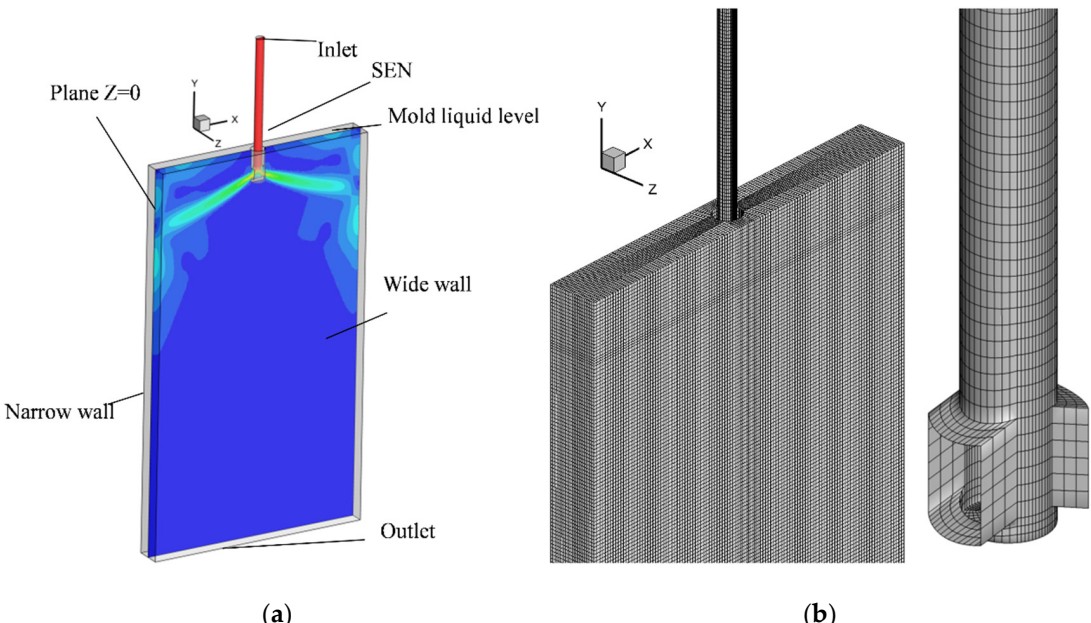

**(a)** **(b)**

**Figure 2.** Schematic diagram of (**a**) computational domain and (**b**) mesh division and local refinement.

**Table 1.** Experiment and simulation parameters.

| Parameters | Values |
|---|---|
| Mold width (mm) | 880, 1050, 1300, 1600, 1800, 2000 |
| Mold thickness (mm) | 200, 237 |
| Steel throughput (ton/min) | 2.2, 3.5 |
| Casting speed (m·min$^{-1}$) | Calculated from the steel throughput |
| Immersion depth of nozzle (mm) | 140, 160 |
| Density of molten steel (kg·m$^{-3}$) | 7020 |
| Viscosity of molten steel (kg·m$^{-1}$·s$^{-1}$) | 0.0055 |
| Argon gas flow rate (L·min$^{-1}$) | 4, 10 |
| Nozzle port size (mm × mm) | 65 × 80, 70 × 90 |
| Nozzle port inclination angle (°) | 15, 20 |
| Bottom shape of nozzle | Concave bottom |
| Argon bubble diameter (mm) | 1 |
| Argon density (kg·m$^{-3}$) | 0.56 |

**Table 2.** Chemical composition of the steel grade (mass%).

| C | Si | Mn | P | S | Al | Nb | Ti |
|---|---|---|---|---|---|---|---|
| 0.002 | 0.005 | 0.66 | 0.039 | 0.01 | 0.042 | 0.006 | 0.0052 |

For the continuous phase of molten steel, the inlet boundary condition of the immersion nozzle was to set the inlet velocity as a fixed value based on the mass balance of inlet and outlet. The inlet velocity was obtained by Equation (17), and the outlet boundary condition at the bottom of the calculation domain was set as the pressure outlet. The boundary conditions of the free surface of the mold top surface were set as stable sliding surfaces, and the boundary conditions of other wall surfaces were set as stable non-slip surfaces.

$$Q = V_{\text{inlet}} \cdot A_{\text{inlet}} = V_{\text{casting}} \cdot A_{\text{outlet}} \tag{17}$$

For discrete-phase argon bubbles, the argon bubble was set to be a rigid sphere with a diameter of 1 mm. The bubbles were all set to escape on the free surface, at the immersion nozzle inlet and mold outlet as their boundary conditions, which means that the discrete phase of bubbles will escape out of the computational domain when they contact these areas. The boundary condition of the wall surface of the immersion nozzle was set as reflection. Under this condition, the bubbles will bounce off the wall with the coefficient of restitution specified by the user. In this paper, the coefficient of recovery was set to be 0.3. The boundary conditions of all other walls were set to capture. When bubbles touch these walls, they will be adsorbed on the walls.

## 4. Results and Discussion

In this paper, the rod deflection method was used to measure the flow velocity of molten steel near the surface of the mold with different widths under the condition of the fixed steel throughput, and the numerical simulation method was used to calculate the flow field in the mold. The influence of the mold width on the flow velocity of molten steel near the surface of the mold, flow field pattern, and turbulent kinetic energy distribution in the mold was investigated.

### 4.1. Influence of Width on Mold Flow Field under Low Fixed Steel Throughput

Under the conditions that the steel throughput was 2.2 ton/min, the argon gas flow rate was 4 L/min, the immersion depth was 140 mm, the submerged entry nozzle port shape was rectangular, and the nozzle inclination angle was 15°. Figure 3 gives the flow velocities near the mold surface with different mold widths under low steel throughput. Figure 3a shows the numerical simulation results of the molten steel flow rate near the mold surface, and Figure 3b shows a comparison between the measurement result and the

numerical simulation result at 1/4 of the width of the mold. If the velocity value was greater than 0 m/s, the molten steel near the mold surface flowed from the narrow wall to the SEN and the flow direction in the mold was the positive velocity. If the velocity value was less than 0 m/s, the molten steel near the mold surface flowed from the SEN to the narrow wall and the flow direction in the mold was the negative velocity [21]. When the mold width was 880 mm, the velocity peak value of molten steel near the mold surface appeared near 1/4 of the width of the mold. When the mold width was 1050 mm, the velocity peak value deviated from 1/4 of the width and approached the narrow wall. When the mold width was 1300 mm, the molten steel velocities value near the surface of the mold were mostly below 0 m/s. Figure 3b shows a comparison between the measured and calculated values of the molten steel flow velocities at 1/4 of the width of the mold. Namely, as the mold width increased, the molten steel flow velocity decreased. When the widths of the mold were 880 mm and 1050 mm, the velocity values were the positive velocity; when the width was 1300 mm, the velocity value was the unstable velocity. The calculated results are in good agreement with the measured results.

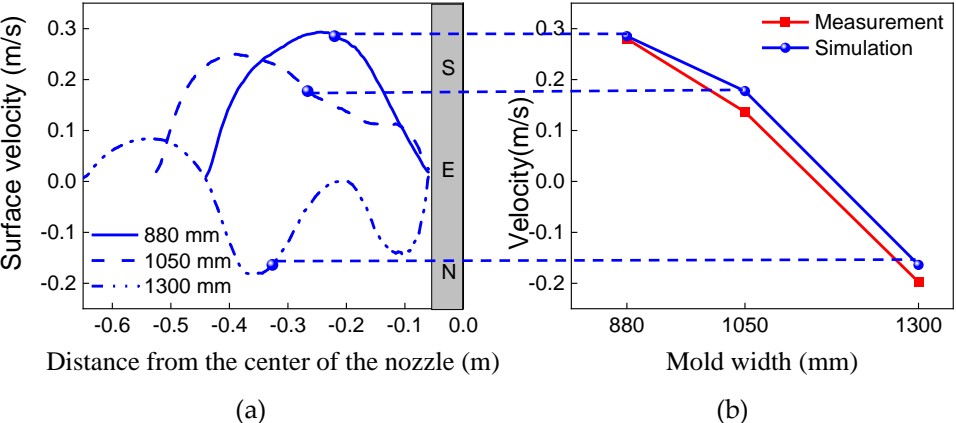

(a)  (b)

**Figure 3.** Flow velocities near the mold surface with different mold widths under the low steel throughput (**a**) distribution in the width direction and (**b**) comparison between measured and calculated values.

Figure 4 shows the velocity contour on the center longitudinal section of the mold with different mold widths under low steel throughput and under the same other conditions as described above. As the width of the mold increased, the velocity of the molten steel in the region of the upper circulation flow in the mold decreased significantly, and the velocity of the molten steel rising up near the nozzle increased obviously. This is because an increase in the mold width leads to an increase in the distance between the side port and the narrow wall of the mold. Therefore, the flow kinetic energy decreases, which directly leads to weakening of the main stream of the upper circulating flow.

Figure 5 shows the streamlines on the center longitudinal section of the mold with different mold widths under low steel throughput and under the same other conditions as described above. It clearly shows that when the widths of the mold were 880 mm and 1050 mm, the flow direction in the mold was the positive velocity. However, when the width of the mold was 1300 mm, the stream A of the molten steel near the SEN wall had a tendency to flow upward and directly impact the top free surface of the mold. Stream B of the molten steel flowed toward the narrow wall. After impinging the narrow wall, it split into an upward flow stream and a downward flow stream. The upward flow stream collided with stream A on the free surface of the mold, which greatly increased the risk of the entrainment of mold flux.

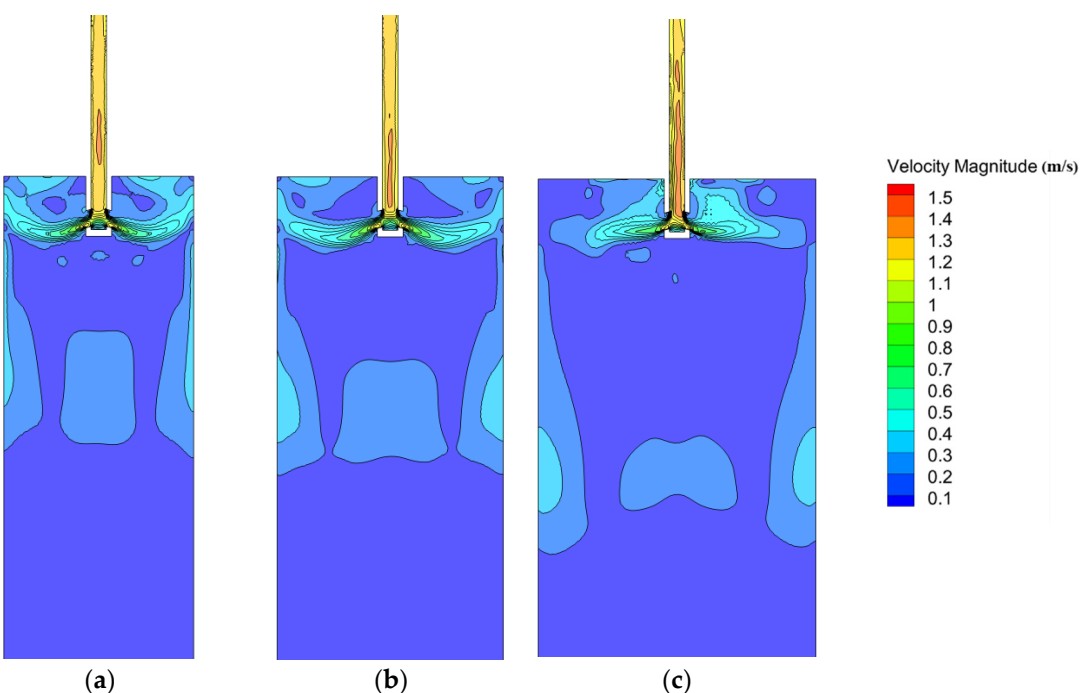

**Figure 4.** Velocity contour on the center longitudinal section of the mold with different mold widths under low steel throughput: (**a**) 880 mm, (**b**) 1050 mm, and (**c**) 1300 mm.

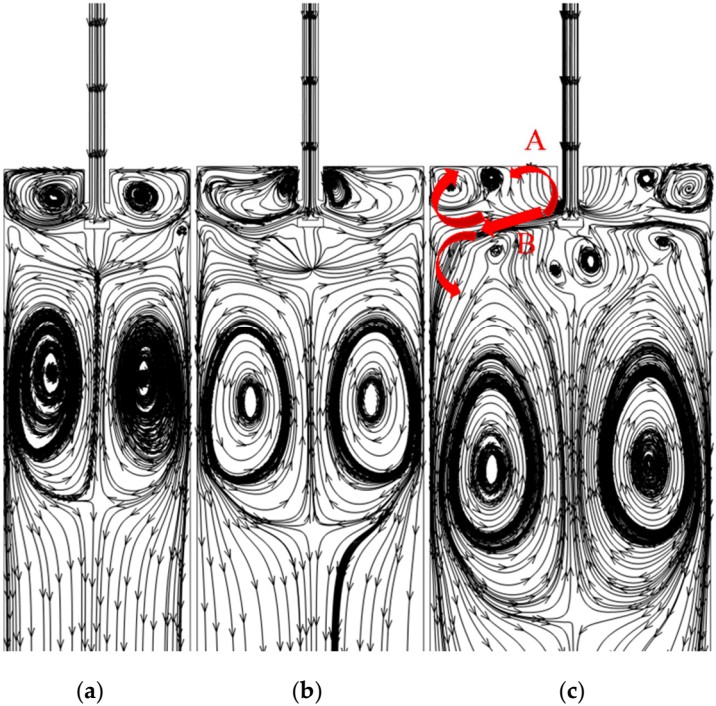

**Figure 5.** Streamlines on the center longitudinal section of the mold with different mold widths under low steel throughput: (**a**) 880 mm, (**b**) 1050 mm, and (**c**) 1300 mm.

Figure 6 shows the contour of turbulent kinetic energy on the free surface of the mold with different widths under low steel throughput and under the same other conditions as described above. When the width of the mold was 880 mm, the turbulent kinetic energy on the free surface was distributed uniformly, and the area with higher turbulent kinetic energy was mainly concentrated at the position of 1/4 of the width of the mold. When the width of the mold was 1050 mm, the turbulent kinetic energy near the nozzle on the

free surface of the mold was significantly higher than that in the other regions. When the mold width further increased to 1300 mm, the high turbulent kinetic energy region occupied the entire area from the SEN wall to over 1/4 of the mold width, and the risk of slag entrainment increased obviously.

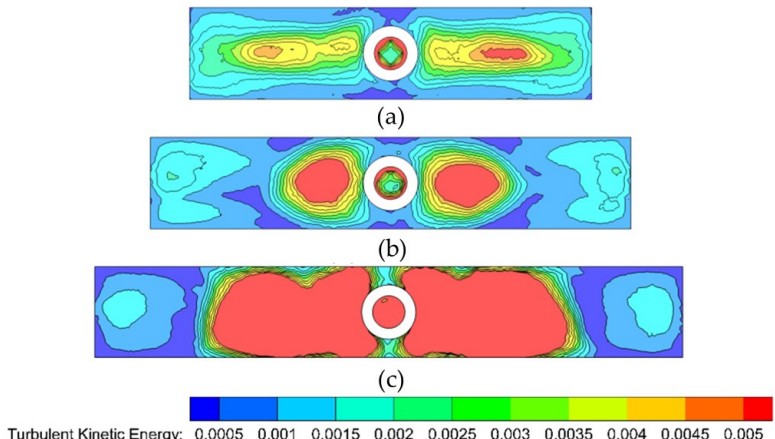

Turbulent Kinetic Energy: 0.0005 0.001 0.0015 0.002 0.0025 0.003 0.0035 0.004 0.0045 0.005

**Figure 6.** Contour of turbulent kinetic energy on the free surface of the mold with different widths under low steel throughput: (**a**) 880 mm, (**b**) 1050 mm, and (**c**) 1300 mm.

The degree of fluctuation of the free surface of molten steel is an important index demonstrating the stability of continuous casting, which is crucial to the surface quality of the automobile exposed panel. The height of the free surface, *H*, is calculated by simple potential energy balance [34]:

$$H = \frac{P_i - P_{\text{mean}}}{\left(\rho_{\text{steel}} - \rho_{\text{slag}}\right) \times \text{g}} \qquad (18)$$

where $P_i$ is the pressure of a certain position on the free surface (N/m$^2$). $P_{\text{mean}}$ is the average value of the pressure on the entire free surface (N/m$^2$). $\rho_{\text{steel}}$ is the density of molten steel (kg/m$^3$). $\rho_{\text{slag}}$ is the density of liquid slag (kg/m$^3$). $g$ is the acceleration of gravity (m/s$^2$).

Figure 7 is the top surface-level profile with different widths under low steel throughput and under the same other conditions as described above. When the width of the mold was 880 mm, the surface fluctuation of the molten steel in the mold was relatively smooth. When the width of the mold was 1050 mm, the fluctuation of the top surface level at the position of 1/4 of the width of the mold increased. When the width of the mold was 1300 mm, the top surface-level fluctuation deteriorated with the position of the peak top surface level changing from 1/4 of the width to the SEN vicinity of the mold, which increased the risk of slag entrainment.

### 4.2. Influence of Width on Mold Flow Field under High Fixed Steel Throughput

Under the conditions that the steel throughput was 3.5 ton/min, the argon gas flow rate was 10 L/min, the immersion depth was 160 mm, the submerged entry nozzle port shape was rectangular, and the nozzle inclination angle was 20°. Figure 8 shows the surface velocity near the mold surface with different mold widths under high fixed steel throughput and a comparison between the measured and calculated values of molten steel flow velocity near the surface of the mold with different widths. Figure 8a shows the numerical simulation results of the molten steel flow rate near the mold surface. As shown in Figure 8a, when the width of the mold was 1600 mm, the surface velocity was greater than 0 m/s, and the velocity distribution in the width direction of the mold had two peaks. The maximum value of the surface velocity appeared between 1/4 of the width of the mold and the narrow wall, and another peak value appeared near the SEN. When the mold

width was 1800 mm, the velocity value of the molten steel at the distance between 0.5 m and 0.9 m from the SEN center was greater than 0 m/s and smaller than 0.2 m/s, indicating that the flow direction of molten steel is from the narrow wall to the SEN. However, the velocity value of molten steel between 0 and 0.5 m from the SEN center was smaller than 0, indicating that the flow direction of the molten steel is from the SEN to the narrow wall. When the width of the mold was 2000 mm, the surface velocity was negative at all points in the width direction, indicating that the direction of the molten steel near the free surface of the mold is all from the SEN to the narrow wall of the mold.

Figure 8b shows a comparison between the measurement result and the numerical simulation result at 1/4 of the width of the mold. As shown in Figure 8b, the simulated results of the molten steel velocity at the 1/4 position of the mold width under the same steel throughout decreased with increasing mold width. However, the measurement results show that the surface flow velocity decreases first and then slightly increases as the width increases. When the width of the mold was 1600 mm, the calculated and measured results of the flow velocity at 1/4 of the surface of the mold were both positive. When the widths were 1800 mm and 2000 mm, both of the measured and calculated results had negative values, which increases the risk of slag entrainment. The calculated results are basically consistent with the measured results.

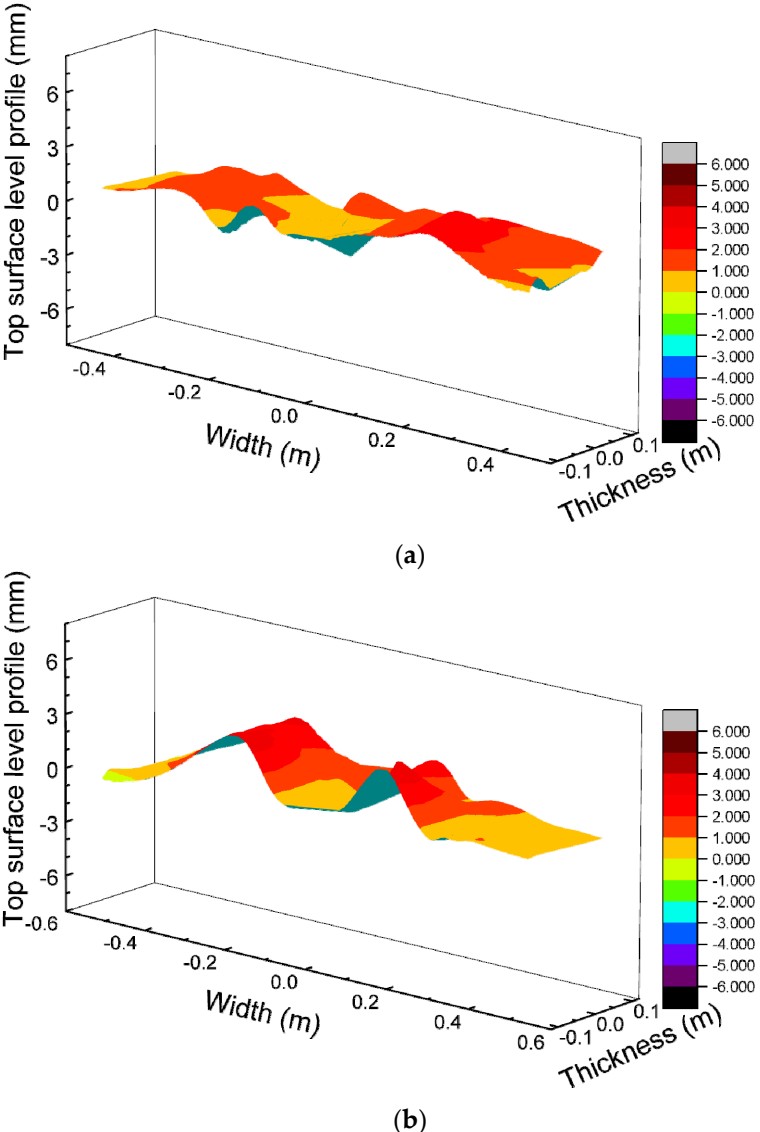

(**a**)

(**b**)

**Figure 7.** *Cont.*

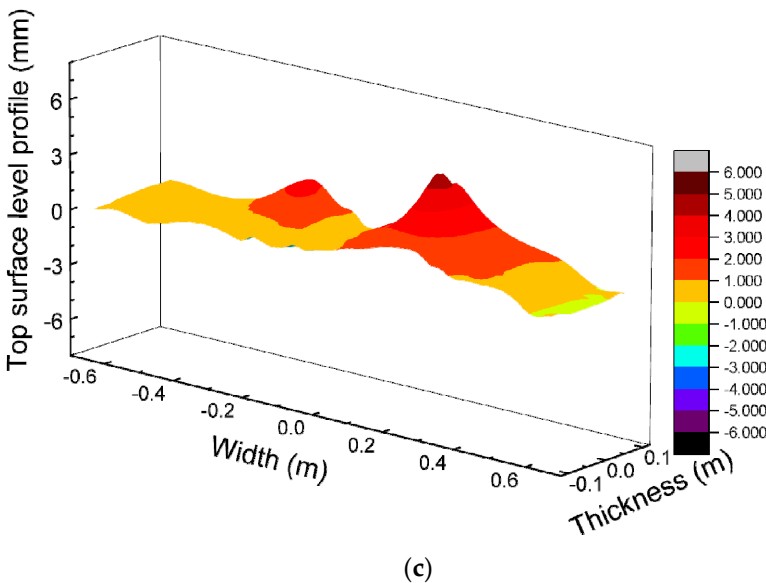

(**c**)

**Figure 7.** Top surface-level profile with different widths under low steel throughput: (**a**) 880 mm, (**b**) 1050 mm, and (**c**) 1300 mm.

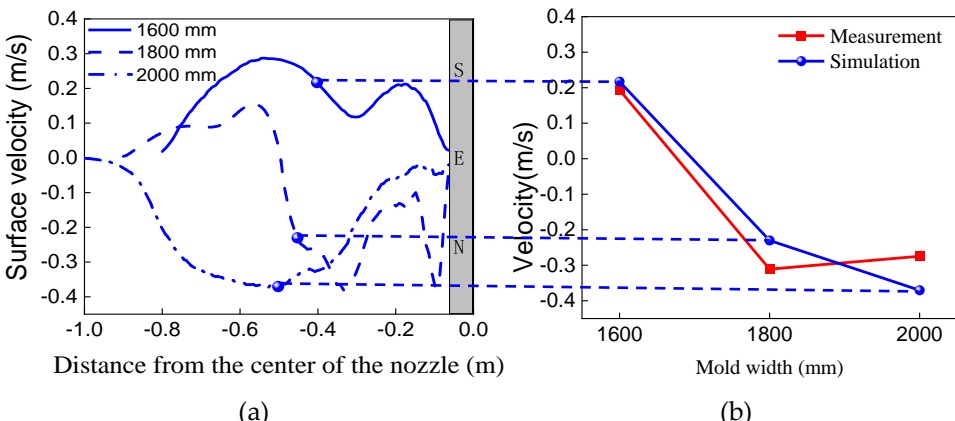

(a)　　　　　　　　　　　　　　(b)

**Figure 8.** Surface velocity near the mold surface with different mold widths under the high fixed steel throughput (**a**) distribution in the width direction and (**b**) comparison between measured and calculated values.

　　　Figure 9 shows the streamlines on the center longitudinal section of the mold with different widths under high fixed steel throughput and under the same other conditions as described above. As the width of the mold increased, the flow field also changed significantly. When the width of the mold was 1600 mm, the flow direction in the mold was a positive velocity. The molten steel flowed out of the side ports of the SEN to the narrow wall. After impacting the narrow wall, it was divided into upper and lower circulating flows to form the positive velocity. Under the action of the rising argon gas bubbles at a gas flow rate of 10 L/min, part of molten steel near the SEN had a tendency to directly flow upward to the top free surface of the mold, but it was suppressed by the molten steel stream flowing up from the narrow wall and along the top free face in the direction to the SEN, as shown in Figure 9a. This also explains why there are two peaks in the distribution curve of the surface velocity in the width direction of the 1600 mm mold in Figure 8a. When the width of the mold was 1800 mm, the flow direction in the mold was the unstable velocity, and the unstable velocity was a flow direction between the positive velocity and the negative velocity [24,27]. The molten steel flowed out from the side ports of the SEN, a part of it flowed to and then impacted the narrow wall to form the upward stream A and

the downward stream B. The surface flow stream C toward the narrow wall met stream A at a distance of 0.5 m from the SEN center to form the unstable velocity. When the width of the mold was 2000 mm, the flow direction in the mold was a typical negative velocity. After the molten steel flowed out from the side ports of the immersion nozzle, it rose upward to the top surface, flowed along the top surface to the narrow wall, and then flowed downward along the narrow wall to form the negative velocity. According to previous research results, the risk of slag entrainment will increase when the flow direction in the mold is the negative velocity and the unstable velocity to increase the incidence ratio of linear defects on the automobile exposed panel [35–37].

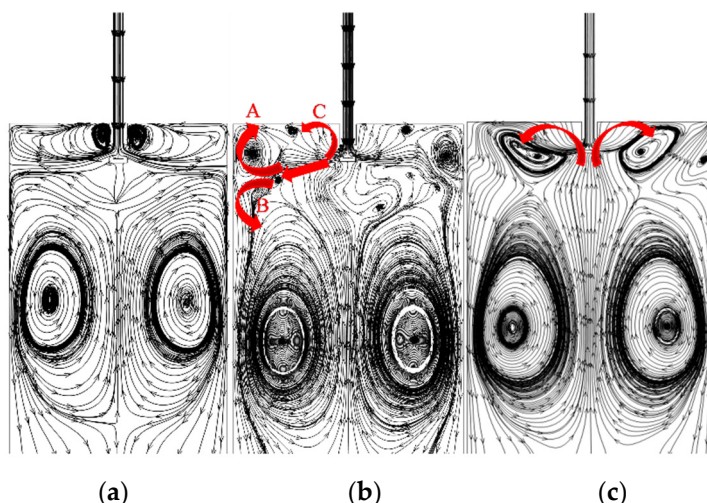

(a)　　　　　　　　　(b)　　　　　　　　　(c)

**Figure 9.** Streamlines on the center longitudinal section of the mold with different widths under high fixed steel throughput: (**a**) 1600 mm, (**b**) 1800 mm, and (**c**) 2000 mm.

Figure 10 shows the turbulent kinetic energy contour on the top surface of the mold with different widths under high fixed steel throughput and under the same other condition as described above. When the width of the mold was 1600 mm, the turbulent kinetic energy near the SEN and the narrow wall was higher than those of other areas, but it was generally at a reasonable level. When the mold widths were 1800 mm and 2000 mm, the high turbulent kinetic energy region near the SEN gradually expanded to the narrow wall. Since the distance of the molten steel flowing from the side ports of the SEN to the narrow wall of the mold was increased, the molten steel stream became weak. Under the action of the rising argon bubbles, the molten steel was easily lifted up to the top surface of the mold. As a result, a high turbulent kinetic energy region appeared in the area near the SEN and expanded to the narrow wall with increasing the mold width, increasing the risk of slag entrainment.

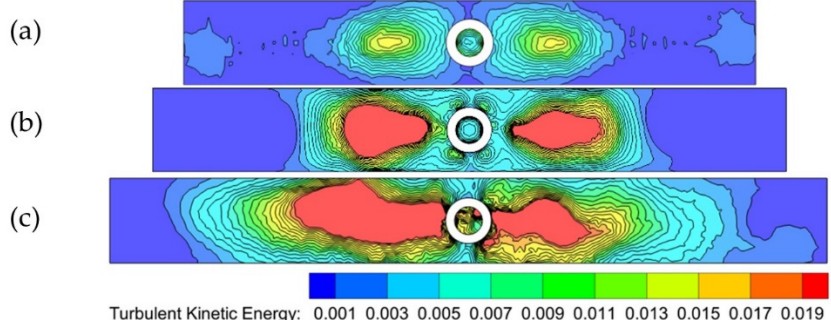

Turbulent Kinetic Energy:　0.001 0.003 0.005 0.007 0.009 0.011 0.013 0.015 0.017 0.019

**Figure 10.** Turbulent kinetic energy contour on the top surface of the mold with different widths under high fixed steel throughput: (**a**) 1600 mm, (**b**) 1800 mm, and (**c**) 2000 mm.

Figure 11 shows the top surface-level profile of the mold with different widths under high fixed steel throughput and under the same other conditions as described above. In addition, under high steel throughput, as the width of the mold increased, the area with the high-level fluctuation gradually shifted from 1/4 of the width of the mold to the vicinity of the SEN. Especially, when the cross-sectional width was 2000 mm, the peak value of the fluctuation near the SEN reached about 7 mm, which easily led to slag entrainment and thus caused surface defects in the automobile exposed panel.

From the above description, it is found that regardless of whether it is low fixed steel throughput or high fixed steel throughput, the change rule of the flow field in the mold with different widths is basically the same. Therefore, in actual production, the argon gas flow rate and the immersion depth of the SEN should be adjusted reasonably to optimize the flow field in the mold with different widths under the same fixed steel throughput.

In fact, the flow field in the slab mold is quite complex, which is affected by a lot of parameters of continuous casting. In our previous papers, the effects of casting speed, argon gas flow rate, immersion depth, and structure of the SEN on the flow field of the slab mold with narrow [15], medium [16], and large [17] widths have been investigated. The coupling of different influencing variables is also important, which we will study in the future.

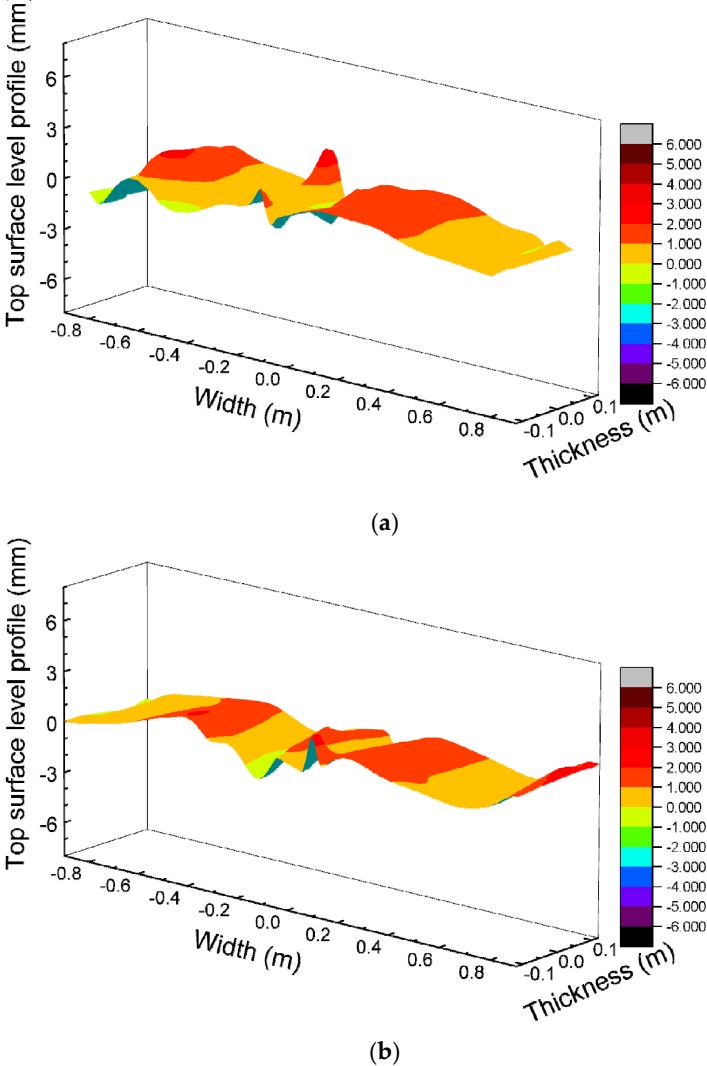

(**a**)

(**b**)

**Figure 11.** *Cont.*

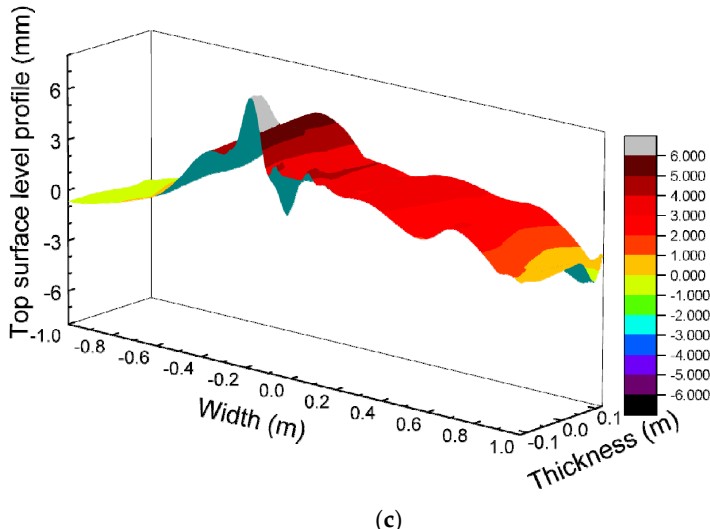

(**c**)

**Figure 11.** Top surface-level profile of the mold with different widths under high fixed steel throughput: (**a**) 1600 mm, (**b**) 1800 mm, and (**c**) 2000 mm.

## 5. Conclusions

In this paper, the rod deflection method was used to measure the flow velocity near the surface of the mold at high temperature, and the standard *k-ε* model coupled with the discrete-phase model (DPM) was used for numerical calculation. The effects of the width of the mold on the surface velocity, flow field pattern, turbulent kinetic energy distribution, and surface-level fluctuation in the mold were studied. The main conclusions are as follows:

(1) Under the conditions of low fixed steel throughput of 2.2 ton/min, a nozzle immersion depth of 140 mm, and an argon gas flow rate of 4 L/min, as the width of the mold increases from 880 mm to 1050 mm and 1300 mm, the flow velocity near the surface of the mold decreases. When the widths of the mold are 880 mm and 1050 mm, the surface velocity is positive, forming the positive velocity. When the width is 1300 mm, the surface velocity is negative, forming the negative velocity. The calculated results are in good agreement with the measured results. The turbulent kinetic energy near the submerged entry nozzle (SEN) gradually increases, and the risk of slag entrainment increases.

(2) Under the conditions that the high fixed steel throughput is 3.5 ton/min, the immersion depth of the SEN is 160 mm, and the argon gas flow rate is 10 L/min, as the width of the mold increases from 1600 mm to 1800 mm and 2000 mm, the velocity near the mold surface decreases. When the width of the mold is 1600 mm, the flow velocity at 1/4 of the surface of the mold is positive. When the widths are 1800 mm and 2000 mm, both surface velocities are negative. The calculated results are basically consistent with the measured results. The high turbulent kinetic energy area near the nozzle shows a trend of expanding to a narrow wall, and the risk of slag entrainment is significantly increased.

(3) Regardless of whether it is a low fixed steel throughput or a high fixed steel throughput, the change rules of the flow field in the mold with different widths are basically the same. Therefore, in actual production, the argon gas flow rate and the immersion depth of the SEN should be adjusted reasonably to optimize the flow field in the mold with different widths under the same fixed steel throughput.

**Author Contributions:** Conceptualization, J.-Q.L. and J.Y.; methodology, J.Y.; software, J.-Q.L.; validation, J.-Q.L.; formal analysis, J.-Q.L.; investigation, J.-Q.L.; resources, C.-L.Z., W.-Y.H., R.-B.J., and Y.-T.G.; data curation, C.M. and Y.G.; writing—original draft preparation, J.-Q.L.; writing—review and editing, J.Y.; visualization, J.-Q.L.; supervision, J.Y.; project administration, J.Y.; funding acquisition, J.Y. All authors have read and agreed to the published version of the manuscript.

**Funding:** The authors gratefully acknowledge the financial support from the Natural Science Foundation of China (No. U1960202).

**Data Availability Statement:** Data supporting reported results can be found in this paper.

**Acknowledgments:** The authors also gratefully acknowledge the financial support from Shougang Jingtang United Steel Co., Ltd. and Tangshan Stainless Steel Co., Ltd.

**Conflicts of Interest:** The authors declare no conflict of interest.

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
