# Peer review of "Effect of Mold Width on the Flow Field in a Slab Continuous-Casting Mold with High-Temperature Velocity Measurement and Numerical Simulation"

_metals, doi:10.3390/met11121943_

Round 1

Reviewer 1 Report

Please follow my comments to revise the manuscript.

Reviewer 2 Report

The paper contains a numerical study of the flow of liquid steel in a continuous casting mold in slab geometry. In parallel, measurements were carried out in an industrial continuous casting plant using a mechanical measuring device to determine the local value of the horizontal velocity near the free surface. Apparently, two cases with different casting speeds are investigated. The focus is on the influence that a variation of the mold width has on the flow structure. Argon is added in the process, probably with the aim of preventing clogging of the Submerged Entry Nozzle (SEN). Gas volume flow and the immersion depth of the SEN increase with the casting speed. Based on the analysis of the flow structure, the authors try to make statements about the risk of slag entrapment at the free surface. 

The flow in a continuous casting mold is highly turbulent and extremely complex. The geometry and a number of process parameters exert a considerable influence on the flow behavior. Since steel quality is highly dependent on the flow in the mold, an in-depth understanding of these relationships is very important for optimization and effective control of the process. For this reason, a large number of studies have already been carried out on this subject in the past. This primarily concerns numerical simulations, but also laboratory-scale experiments and industrial trials. 

Against this background, it must be stated that the present study unfortunately does not make any significant contribution to a better understanding of mold flow and its quality clearly falls short of already known publications. I do not deny that the results presented here may be of interest to the operator of the industrial continuous casting plant, but there is no gain in knowledge for the scientific community.  Obvious questions are not addressed and adequately discussed. The results are not compared and evaluated with similar studies in the literature. Therefore, I see no need or benefit for this work to be published and cannot recommend publication in the journal "Metals".

In detail, this is due to the following reasons:

The properties of the flow in the mold depend on a number of factors and ultimately the respective situation can only be evaluated in the context of all influencing variables. However, the focus of this study is only on the effect of a variation of the mold width. Other factors, e.g. Ar bubbles, are not discussed and evaluated. Discussion and interpretation of the results are very poor. They are not really surprising and also not new. 

It is to be expected that when the mold width is increased, the intensity of the impact of the jet on the side wall is reduced. As a result, the jet is no longer split so strongly and is partially deflected upward. As a result, the upper rolls above the nozzle ports are less pronounced. This phenomenon is already sufficiently known from previous studies. However, in contrast to the discussion in Chapter 4, I do not see a transition to a single-roll structure anywhere. Figure 5 shows the decay of the upper roll into a multi-roll structure at the surface. In Figure 9, the multi-roll structure is seen at an intermediate mold width, while again at the widest width, a double-roll structure apparently reappears. Why is this so? In general, it is difficult to judge to what extent the results inspire confidence, since some important information is missing, e.g. discretization (mesh size).

The code was apparently only validated on the measurement results. However, this is very questionable and in reality cannot be considered as validation, since from the experimental side only one local measuring point per parameter combination is available. In addition, the measurement principle used is very coarse and inaccurate and quite error-prone and thus not suitable for a proper validation of the numerics. Analysis and discussion of measurement errors is completely missing, e.g. how stable is shape of rod? Does material dissolve after a certain time or can material also settle there because temperature is lower? This has been investigated in detail e.g. when using so-called nailboards to measure near-surface flow. However, all these studies are not cited here.

Contrary to what is claimed in the introduction, the measurement technique was not originally developed by the authors. The measurement principle has been known for decades under the name reaction probes, see for example Szekely et al. Metall. Trans. B (1977), El-Kaddah et al. Metall. Trans. (1984), Argyropoulos, Scand. J. Metall.(2000).

Another proven method for gaining knowledge is the use of physical models. These are largely operated with water. For some years, however, there have also been powerful experimental facilities with liquid metals. The advantage of these experiments on a laboratory scale is that they can really be used to obtain sufficiently accurate measurement data to validate numerical codes. However, nothing of this is mentioned in the paper. Corresponding references are completely missing. 

What is the role of two-phase flow and gas distribution? It is well known that the assumption of 1mm bubbles is quite unrealistic. Bubbles are usually larger and thus deformable. Interactions such as bubble coalescence are likely to occur to a large extent and are essential to the bubble size that is established and consequently to the ascent behavior and feedback on the metal flow. The two-phase model used by the authors is thus far too simplistic and is unlikely to lead to realistic results.

Do the plots in Figure 7 show snapshots of the deflections of the free surface? What are the dynamics given the large deflection? Do surface waves occur?

The quality of the illustrations generally needs to be improved. What do figures 3 and 8 actually show in detail? I assume the lines are the numerical results? Clear labels are missing.

The paper talks about "narrow surface". I assume that we are talking about the "narrow wall".

Reviewer 3 Report

Dear authors,

I found your article interesting and generally well-written. However, I have some issues with various matters. Please consider them and make necessary improvements. After that, I am positive to publish the article without a second review.

  1. I wonder about the type of article. The title is quite general but the authors from very beginning of the Introduction claim this is rather a case study for the “automobile exposed panel”. I suggest clarifying that perhaps in the title and/or in the Abstract, too.
  2. In Figure 4 it would be good to add the velocity measuring unit.
  3. Table 1: what is the material of the mold and what is the grade of the steel?
  4. Table 1: Argon bubble diameter is 1mm for modeling purposes. How uniform the size of the bubbles is? Any data and/or idea?
  5. What software has been used for modeling? Commercial one or own code?
  6. Page 11: the formula should be continuously numbered as (18).

Sincerely,

Reviewer

Round 2

Reviewer 2 Report

In the first round of peer review, I did not recommend the submitted manuscript for publication because, on the one hand, I am not convinced of the originality of the results and, on the other hand, I do not consider their importance for scientific knowledge to be sufficient for publication. In addition, there are some doubts about the appropriateness of the methods used, the validity of certain assumptions, and the confidence in the measurements and numerical results.

The authors have now submitted a revised version within about two weeks, although one has to wonder whether a substantial rewriting of the manuscript is possible in this short time.  Accordingly, it can be stated that the fundamental objections have not been completely resolved. The authors' responses to my comments didn't really convince me either.

Thus, in response to the objection that the study does not contain enough new information that would be really important for the scientific community, no corresponding evidence was provided, but it was simply asserted that it is so.

Furthermore, I miss a comprehensive consideration of the influence of different parameters on the properties of the flow.  In this respect, the authors only point out that they had considered the effect of other parameters more closely in earlier studies. However, the complexity of the problem due to the coupling of different influencing variables is disregarded. A least a respective discussion is missing in the manuscript under consideration.

The discussion regarding the occurrence of a single-rool structure is not convincing. he structure i Fig. 9c is not a single-roll, but a double-roll structure in which the upper rolls are only very weakly developed. The flow pattern is still symmetric. The natural cause of a single-roll flow asymmetry in the mold could be an asymmetric clogging of the SEN, transient flow oscillations induced by strong gas injection, etc. The authors should have a closer look into the literature (for instance Zhang et al, MMTB, 38, 2007).

The authors cite patent applications to prove that at least some parts of the measuring apparatus used were developed by themselves. But that is not what I mean, the underlying measuring principle is already quite old, and it would only be fair to point out these precursor works. Apart from that, the measurement method is rather crude and provides only inaccurate estimates of the flow velocity, which in my view is not sufficient to sufficiently validate a numerical code. a satisfying error discussion is missing.

This is the advantage of the physical models. One can make much more accurate measurements and generate appropriate data sets for validation. It is good to hear that the authors want to work with such a model. However, the task for the present manuscript is to mention and appreciate the previous work with physical models accordingly and to discuss previous results in the light of the results known from the models. 

It is still not clear to me where the authors get the assumption that the bubbles in the steel are smaller than 2.2 or 3.1mm. This is not correct and corresponding studies that have investigated this in more detail so far are not cited and discussed here. I note that the authors did not understand how the bubbles form in the nozzle and that the numerical simulations are likely to give wrong results because the bubble sizes are assumed to be too small.

An analysis of the surface deflections (Figure 7) based on a few snapshots is insufficient and provides little insight into the physical mechanisms. 

All things considered, I must insist on my original assessment of the present manuscript.
